# Emerging Treatments for Disorders of Consciousness in Paediatric Age

**DOI:** 10.3390/brainsci12020198

**Published:** 2022-01-31

**Authors:** Hassna Irzan, Marco Pozzi, Nino Chikhladze, Serghei Cebanu, Artashes Tadevosyan, Cornelia Calcii, Alexander Tsiskaridze, Andrew Melbourne, Sandra Strazzer, Marc Modat, Erika Molteni

**Affiliations:** 1School of Biomedical Engineering & Imaging Sciences, King’s College London, London WC2R 2LS, UK; hassna.irzan@kcl.ac.uk (H.I.); andrew.melbourne@kcl.ac.uk (A.M.); marc.modat@kcl.ac.uk (M.M.); 2Department of Medical Physics and Biomedical Engineering, University College London, London WC1E 7JE, UK; 3Scientific Institute IRCCS E. Medea, Acquired Brain Injury Unit, 22040 Bosisio Parini, Italy; marco.pozzi@lanostrafamiglia.it (M.P.); sandra.strazzer@lanostrafamiglia.it (S.S.); 4Faculty of Medicine, Ivane Javakhishvili Tbilisi State University, Tbilisi 0179, Georgia; nino.chikhladze@tsu.ge (N.C.); alexander.tsiskaridze@tsu.ge (A.T.); 5Faculty of Medicine, Nicolae Testemitanu State University of Medicine and Pharmacy, MD-2004 Chišināu, Moldova; serghei.cebanu@usmf.md (S.C.); cornelia.calcii@usmf.md (C.C.); 6Department of Public Health and Healthcare Organization, Yerevan State Medical University, Yerevan 0025, Armenia; artashes.tadevosyan@meduni.am; 7Rehabilitation Service, “Usratuna” Health and Rehabilitation Centre, Juba, South Sudan

**Keywords:** Paediatric Disorder of Consciousness, treatment of Disorder of Consciousness, pharmacology of paediatric brain injury, rehabilitation of paediatric brain injury, children’s brain injury global health

## Abstract

The number of paediatric patients living with a prolonged Disorder of Consciousness (DoC) is growing in high-income countries, thanks to substantial improvement in intensive care. Life expectancy is extending due to the clinical and nursing management achievements of chronic phase needs, including infections. However, long-known pharmacological therapies such as amantadine and zolpidem, as well as novel instrumental approaches using direct current stimulation and, more recently, stem cell transplantation, are applied in the absence of large paediatric clinical trials and rigorous age-balanced and dose-escalated validations. With evidence building up mainly through case reports and observational studies, there is a need for well-designed paediatric clinical trials and specific research on 0–4-year-old children. At such an early age, assessing residual and recovered abilities is most challenging due to the early developmental stage, incompletely learnt motor and cognitive skills, and unreliable communication; treatment options are also less explored in early age. In middle-income countries, the lack of rehabilitation services and professionals focusing on paediatric age hampers the overall good assistance provision. Young and fast-evolving health insurance systems prevent universal access to chronic care in some countries. In low-income countries, rescue networks are often inadequate, and there is a lack of specialised and intensive care, difficulty in providing specific pharmaceuticals, and lower compliance to intensive care hygiene standards. Despite this, paediatric cases with DoC are reported, albeit in fewer numbers than in countries with better-resourced healthcare systems. For patients with a poor prospect of recovery, withdrawal of care is inhomogeneous across countries and still heavily conditioned by treatment costs as well as ethical and cultural factors, rather than reliant on protocols for assessment and standardised treatments. In summary, there is a strong call for multicentric, international, and global health initiatives on DoC to devote resources to the paediatric age, as there is now scope for funders to invest in themes specific to DoC affecting the early years of the life course.

## 1. Introduction

Disorders of Consciousness (DoCs, Table 1) represent a small but expanding spectrum of rare conditions receiving growing attention due to the associated clinical, emotional, and economic burdens. The number of children with DoC is increasing [1], as survival and disease prevalence rise. However, the incidence and prevalence of DoC fluctuate between countries, due to diverse socio-cultural factors, health policies, and hygiene standards [2]. Furthermore, early withdrawal of care can contribute reducing prevalence in some countries [2]. Importantly, more than 62% of children with Unresponsive Wakefulness Syndrome (UWS, also termed vegetative state (VS)) regain consciousness within the first year post-injury [3], and 27.8% of those with UWS and in a Minimally Conscious State (MCS) have disease duration longer than six years [4], supporting the notion that younger age is generally associated with greater chances of better outcome compared to adulthood [5]. Nevertheless, the chances of poor outcome are still high [6]. It is thus imperative for the scientific community to anticipate and safely explore the emerging treatments for DoC in paediatric age.

Traumatic brain injury (TBI) is still one of the leading causes of paediatric DoC (~47.8% [10]). Although TBI is declining in Europe and North America [11,12,13], low- and low-middle-income countries (LMICs) see a worrying increase due to growing motorisation, affecting especially children in South Asia and the West Pacific [14,15,16]. Falls and violence are major concerns for youth in war conflict areas [14]. Infections are another leading cause of DoC in children <5 years [17], with particularly high incidence in sub-Saharan Africa and South Asia [18].

As the causes of DoC are variegated, and the disability often severe and prolonged, the treatment course and outcome are highly variable. The assessment of consciousness must be performed through standard tools, specifically validated or adapted for paediatrics (i.e., CNCS, LOCFAS, and PALOC instruments) [10,19]. Of note, the Coma Recovery Scale—Revised (CRS-R) [20], which is the standard in adults, has been increasingly used in paediatric research (e.g., [21,22,23,24]), but, to date, it has only been tested in healthy children, and not yet in children with DoC [25]. Inappropriate use of non-paediatric scales can result in motor or cognitive requests beyond the child’s development stage and behavioural capabilities. This causes underestimation effects at best, or introduces saturation in measurements, i.e., flooring or ceiling effects, thus distorting or impeding any mapping of the patients’ evolution over time, especially in correspondence with treatments.

A wide range of therapies is available to reduce DoC-related complications (e.g., epilepsy). In particular, spasticity and dystonia can hamper timely and appropriate delivery of rehabilitation and induce pain, and thus need to be treated. General guidelines exist, which are not specific to children, although they partially target paediatric needs [26,27]. On the other hand, medications for spasticity can induce fatigue as a side effect, thus resulting in a disadvantageous factor to the overall regaining of consciousness and dragging patients into a vicious circle.

In addition to this, no ultimate treatment for consciousness is available. To date, the only therapy recommended in clinical guidelines for treating DoC is amantadine [28], but evidence from children is low. Amantadine, when prescribed, is prevalently administered after severe TBI in paediatrics, and is effective in only ~55% of cases. There are no guidelines specific to the treatment of paediatric DoC [28]. For this reason, there is a critical need to develop highly evidence-based interventions to promote responsiveness, improve rehabilitation, and aid treatment decision-making in children.

## 2. Neurorehabilitation

Intensive neurorehabilitation is the elective therapy in paediatric DoC. Due to the complexity of the disease, it is commonly delivered in a multidisciplinary approach encompassing physical, oral-motor, occupational therapy, and neuropsychological interventions [29].

**Why and to whom** to deliver neurorehabilitation? Neurorehabilitation mainly aims at the *habilitation* of physiological functions otherwise compromised by the disease, and relies on training through physiological processes such as repetition and (re)learning [30,31]. For these reasons, it has very few side effects, it can be applied to all ages from birth, and its principles are not specific to a pathology. A long clinical tradition makes neurorehabilitation techniques generally highly documented. Conversely, the limitations of neurorehabilitation are that functional targets can be non-specific, the treatment itself can be highly subjective to the therapist expertise and approach, demonstration of efficacy can be arduous, and precise treatment documentation can be burdensome.

**What** is the aim of neurorehabilitation? (Multi)sensory stimulation is at the core of early intervention. It aims to generate a comprehensive set of environmental prompts by stimulating all the five senses and the motor system [32,33], and it is deemed to favour the re-organisation and the re-building of the behavioural repertoire [34]. The ultimate goal is to leverage residual activation to—at least ideally—reach out to the entire brain network and foster functioning in the damaged neural tissue through *facilitation* [35].

**When** to deliver neurorehabilitation? Maximal effectiveness is obtained when the intervention is administered early after the injury, when sufficient stability of the vital parameters, resolution of fractures, and respiratory capacity allow safe treatment delivery [31,36]. Most recent views set the beginning of rehabilitation as during the in-stay in intensive care units, acknowledging that the intervention needs to be modulated according to the patient’s own clinical needs [37]. In addition, a critical timeframe may exist, beyond which a treatment becomes less effective. However, endotypes have been identified that achieve slow but steady progress over several years [38].

**How** to deliver neurorehabilitation? Current research predicates the use of salient stimuli (i.e., events with an emotional or affective valence or content) [22,24]. In particular, stimulus personalisation using the parent’s voice and touch [24], the live performance of preferred lullabies [21], the interaction with (household) pets [39], and themes matching pre-injury hobbies and inclinations [40] have been proven effective. In addition, the repetition of stimulation to match the patient’s optimal arousal is also recommended to maximise responsivity [22]. However, comparative trials for testing the efficacy of different stimulation techniques are missing, as well as studies escalating the number and frequency of sessions to systematically find the best treatment intensity.

Recent advances have focused on the design of the Rehabilitation Treatment Specification System (RTSS) [41,42], a conceptual framework that defines elements of therapy as “ingredients of rehabilitation” and links each of these elements to the improvement of patients’ specific functions. To standardise the reporting, the International Classification of Functioning, Disability and Health—version for Children and Youth (ICF-CY) is recommended [43]. The joint use of these theoretical frameworks represents a paradigm shift, moving from the prescription and reporting of treatment within a broad discipline (e.g., 5 min sensory stimulation) to the administration of the specific therapeutic element (e.g., 20 guided hand touches of different textures) to target the corresponding physiological function (e.g., hand perceptual increase). The next impelling step is the extension of this framework to the patient’s assessment, by linking the content and material of the treatment specification systems to the metrics and possibly the tools for standard assessment of paediatric DoC.

The well-established neurorehabilitation approach is increasingly integrated with emerging pharmacologic and technology-assisted interventions. Moreover, global efforts are being put in place towards harmonised data collection, the establishment of large databases, and the deployment of data-driven methods and artificial intelligence. This is expanded in the following sections.

## 3. Pharmacologic and Regenerative Therapies

Pharmacological treatments proposed for DoC target heterogeneous aspects: awareness, consciousness, responsivity, brain connectivity, sleep re-structuring, neuronal survival, and function. However, none are supported by substantial evidence, as rigorous clinical trials are generally lacking. In children, no reliable trials have ever been conducted. In adults, only one trial on amantadine was methodologically sound, although affected by limitations [44]; the main outcome measured neurological recovery instead of consciousness, amantadine lost efficacy at post-treatment follow-up, and narcotics were used more in the placebo group. Recent reviews provide a comprehensive description of other disparate treatments tested for DoC (Appendix A) [45,46,47].

Considering the very heterogeneous drug repertoire and the inconsistency of trial results, a conceptual systematisation of current research questions may be useful to promote methodological improvements.

**Why** should drugs be used? The most principled pharmacological intervention for DoC involves the anterior forebrain mesocircuit [48]. Monoamine boosting by any means involving antidepressants, serotoninergic agonist psychedelics, stimulants, or dopaminergic agents, is believed to promote the recovery of function in the mesocircuit [49,50]. GABAergic benzodiazepines and Z-drugs may inhibit neurons in the globus pallidus, relieving inhibition from the mesocircuit [51]. Other treatment approaches promote arousal by the orexinergic [52] or melatoninergic [53,54] systems, as well as brain regeneration by nerve growth factor [55], cerebrolysin [56], or diverse cell transplantations [57,58].

The existence of heterogeneous and not fully clear hypotheses underlying DoC, and therefore the identification of such diverse therapeutic aims, has led to the testing of variegated treatments and the generation of scattered evidence. There is still uncertainty about the precise match between therapeutic aims and therapies, as there is no treatment, single or combined, for DoC as a whole. Moreover, there is no guarantee that even great efficacy in one aspect of DoC (mesocircuit, arousal, brain regeneration) would allow consciousness recovery. Future studies should identify therapeutic targets first, and then consider drug candidates based on the chosen target.

Complementary to the stimulation of consciousness, the promotion of sleep restructuring and circadian alternation are concurrently addressed in paediatric DoC, using a largely overlapping pool of pharmacological interventions and a timed approach. No specific recommendations exist for the management of circadian re-organisation in children with DoC; however, recent guidelines on the management of sleep in neurological diseases, and severe brain injury in particular, can give precious indications [59,60,61].

**What** is drug efficacy? Investigation in this field has frequently produced data on outcomes different from consciousness itself (e.g., on neurological impairment), inadequate statistical power, or biased expectations [62,63]. Future studies need to anticipate the therapeutic aim and standardise outcomes (e.g., measures of mesocircuit function, arousal, etc). Standard outcome measures allow result comparisons and meta-analyses, which may surpass the limitations of small sample sizes.

**Who** should be treated with drugs? Brain injury severity and aetiology should be considered when assembling cohorts for trials on DoC. Neuroimaging techniques can be used to detail the damaged brain areas and functions and define where to expect drug effects [64]. Patient age may be a crucial factor to consider, especially with respect to regenerative medicine approaches involving growth factors and cell transplants. Under a principle of precaution, growth factor and stem cell treatments raise greater safety concerns for young patients [65], including those regarding unpredictable effects on brain plasticity.

**When** should DoCs be treated with drugs? The heterogeneity of time from injury to treatment should be reduced in future studies. The recovery phase from injury should be described and considered as a confounding variable. Aspects that may change substantially throughout recovery include the level of neuroinflammation [66], the loss of neuronal function in specific areas, and the possible brain remodelling that has already occurred [67]. Most drug treatments may be helpful only in limited windows of opportunity, and data are needed in this regard.

**How** should drugs be used? Data on drug dosing are missing. Drug distribution and pharmacokinetics may differ in damaged, as compared to intact, brains [68,69]. Drug safety and overdosing may be difficult to monitor due to the lack of patients’ responsivity. Polytherapy [70] and multiple drug interactions may complicate pharmacokinetics and safety. In this regard, it is worth mentioning that benzodiazepine or Z-drug overdosing would pose much lower threats as compared to, for instance, monoaminergic hyper-stimulation, which is known to trigger excitotoxicity [71]. An open issue is whether treatment of DoC should be prioritised over concurrent neurological or internal medicine issues, for instance, when using antiepileptic drugs with known adverse effects on cognition or central muscle relaxant drugs known to be sedative.

At present, no treatment recommendation can be made regarding adult and, to a greater extent, paediatric patients. It must be stressed that producing more low-quality evidence would just add to the confusion. Drug treatments for DoC are still in such a preliminary phase that therapeutic targets must first be validated, and this should be done with preclinical models to accelerate drug positioning. From a clinical perspective, better integration of neuropsychological assessments, imaging techniques, and clinical pharmacology is required to move forward and enable the identification of promising single or combined therapeutic strategies.

## 4. Medical Technologies for Treatment

A small subset of therapeutic principles is appealing to the field of paediatric DoC, although at different stages of development. Guided by both precautionary principles and the need for design and ergonomic refinement, technologies are commonly tested and validated on adults first, and then adapted to children.

**Why** use technology? Rehabilitation is increasingly based on technology, which gives a chance of treatment objectivation and standardisation while preserving therapists’ freedom to choose activities and supervise.

**What** are the technological approaches?

### 4.1. Neurostimulation

We anticipated that multisensory stimulation is the elective non-pharmacological treatment in paediatric DoC. Sensorimotor stimulation is the first candidate for delivery via devices. Instrumental early mobilisation [72] of the patient serves the twofold purpose of avoiding immobility due to long bedding periods and providing neuromotor stimulation, thus improving DoC outcome [31]. This can be safely and easily achieved through tilting mechanics or shape memory alloys [72]. Repetitive Nerve Stimulation aims at providing electrical stimulation to a primary nerve afferent to the brainstem (and cortex) (Figure 1A) [73,74]. A preliminary report on one child [32] showed both the short- and long-term positive effects on functional brain connectivity.

### 4.2. Neuromodulation

Low-intensity electromagnetic treatment of the skull aims at manipulating either the *threshold* for or the *frequency* of the neural action potential firing in the cortex underneath. Appropriately prolonged manipulation is deemed to modulate the cortical excitability, condition the long-range and inter-hemispheric brain connectivity [75], and induce plasticity, thus promoting the subject’s behavioural responsiveness and improvement [76,77,78,79] (Figure 1B). The use of Deep Brain Stimulation, the invasive predecessor [80], is discouraged due to the risk of severe or catastrophic complications, and after cases of device removal were reported [81]. However, the non-invasive counterpart, referred to as transcranial Electrical Stimulation (tES), has provided a proof-of-concept for improvement of function in children with DoC [82].

In principle, neuromodulation can be employed either as treatment per se or for priming the brain to enhance response to concurrent pharmacological or neuro-rehabilitative treatment [83]. The latter option has already been explored in paediatric motor disorders [84], with conflicting results [85], for reinforcing (physical) therapies aiming to reduce specific symptoms.

### 4.3. (Targeted) Drug Delivery

One appealing prospect is the localised, targeted, and minimally invasive delivery of tuneable doses of therapeutics to specific anatomical structures, sparing untargeted tissues. Focused ultrasounds [86] allow the transient and localised opening of the blood–brain barrier in a graded manner for targeted chemotherapy and delivery of neurotrophic factors to the central nervous system (CNS) (Figure 1C). Research on the so-called *millirobots* promises controlled delivery of payloads to target tissues [87] through soft paramagnetic droplet carriers manipulated by external magnetic fields (Figure 1D). These are able to direct the delivery of cellular, molecular, and protein therapies to specific regions of the CNS, even against fluid flows [88]. These techniques, albeit in their infancy, have the potential to reduce the overall dosage, increase therapeutic efficacy via targeted delivery and enhanced retention, and decrease side effects by minimising off-target deposition, tissue absorption, and systemic toxicity (if any).

**Who** can benefit from these treatments? Although all three approaches raise interest in the field of paediatric DoC, some exclusion criteria apply to each technique (Table 2). *Neurostimulation* has specific ethical implications for patients unable to communicate pain. Clinical effects of *neuromodulation* in adults with DoC are promising but heterogeneous (~40% of adults in MCS and ~10% in a vs. improved signs of consciousness [77,78,89,90,91,92] with transcranial direct current stimulation; safety in children is still under investigation [93]). Overall, a one-size-fits-all treatment is unlikely, and we envisage that well-defined coma endotypes will receive tailored treatments in the future.

On a side note, *neuromodulation* is also deployed for chronic pain treatment. Palliative care is probably less explored than consciousness recovery in children with DoC, although it is at least equally important in practice. *Neuromodulation* has shown potentiality for pain treatment in adults [96], provided that accurate pain assessment is applicable to the subject. In children with DoC, the pain assessment remains arduous; however, an appropriate scale exists for children aged three years and older: the Non-Communicating Children’s Pain Checklist—Postoperative version (NCCPC-PV) [97,98]. Some scales for consciousness employed in paediatrics such as the LOCFAS and CNCS inherently contain one or more pain items, as well as the CRS-R.

**When** to start treatments? *Neurostimulation* is generally the earliest treatment administered along the recovery path, especially when conducted through non-invasive techniques such as the patient’s instrumental mobilisation. When applying *neuromodulation,* the extent of the functional reorganisation in the brain depends, at least to some degree, on the patient’s active participation in treatments. For this reason, *neuromodulation*, combined with manual or robotic physiotherapy and cognitive treatments, is generally proposed later along the recovery course [47]. However, no indication exists on the timeline for administering *neuromodulation* in children with DoC, as the associated mechanisms of neuronal potentiation and repair are still largely unknown.

**How** to make these technologies applicable? From an engineering standpoint, paediatric applications often impose additional challenges such as miniaturisation, component downsizing, weight reductions, and imposition of smaller but more precise forces and torques [99]. If electromagnetic energy is applied, intensity reductions might be needed, and on-purpose dose-scaled clinical trials or simulations could be required at additional costs. In addition, indications for the ethical use of medication for neuroenhancement in paediatrics are available [100], but they are specific neither to instrumental stimulation nor to DoC patients.

## 5. Challenges and Opportunities in Low- and Middle-Income Countries

Low- and Middle-Income Countries (LMIC) have a higher incidence of acquired brain injuries, with TBI twice as common as in High-Income Countries (HIC) and affecting ages younger on average [101]. Common causes are poorer conditions and lower safety standards of roads, including old or insufficient infrastructure; high rate of infections and cerebral malaria in children who tend to be malnourished; war conflicts; governmental neglect; and urban-centred services, leaving rural or remote areas underserved. Defective emergency transport and technological obsolescence negatively impact trauma survival, severity, and outcome [102]. Children pay the highest toll in many LMICs, due to poorer socioeconomic conditions affecting parenting and availability of home supervision and causing child labour, sometimes in unsafe or unhealthy environments. Other causes are the lack of professional training on paediatric trauma care, less specialised paediatric intensive care units, lack of age- or size-appropriate equipment, and underestimation of the long-term effects, which leaves the milder cases of TBI unreported and untreated [14].

However, for patients with DoC living in LMICs, the chronic lack of post-acute services is the real plague [103]. Rehabilitation is often inaccessible to the wider population, as governmental rehabilitative services are absent or insufficient. Typically, the national healthcare systems (where existing) and health insurance do not cover treatments beyond acute care, in contexts of widespread household poverty or economic fragility. This leaves the patients to bear the cost of a privately led and poorly quality-assessed rehabilitation offer, unless non-governmental organisations (NGOs) take charge of this gap. Intuitively, this practice generates greater discrimination in countries with variegated socioeconomic backgrounds, where ethnicities with higher economic capacity show better clinical outcomes overall [102].

However, LMICs can, in many cases, leverage favourable factors. In some contexts, prolific families, wide family and community connections, and religious beliefs can provide patients with a large support network beyond close relatives. In addition, countries with political stability and young demography are seeing unprecedented student access to university education in healthcare professions, ensuring a new generation of professionals and creating demand for continuous professional education (CPE) programs.

## 6. The Way Forward

From a worldwide perspective, improvement of treatments for paediatric DoC requires distinct actions, depending on contexts, resources, and primary needs.

In LMICs, investing in education is crucial to improving healthcare, as increased literacy enables demand for safer infrastructure and higher-quality health services and raises awareness about preventive healthcare measures. Focus should be on both the quality and the quantity of healthcare services; this involves better organisation, boosting the number and preparedness of specialised healthcare professionals, building suitable medical infrastructure, and implementing welfare-based healthcare systems.

In all countries, and especially where care is more advanced, investing in research and setting standards for well-justified treatments for paediatric DoC is crucial. The research literature on paediatric DoC is mainly formed by small-sized studies, which are observational in nature, and the reporting of heterogeneous outcome measures. The setup of larger interventional trials is limited by the cautionary principle of safety and tolerability in children, which very often considers, in the absence of adequate preliminary data, the potential life-long effects of the intervention on the developing and plastic brain. A combination of good practices can help overcome this paradox of the ‘safeguarded children’ and increase the evidence in the field.

### 6.1. Standardisation and/or Protocol Adaptation

Although the prospective recruitment of large cohorts is often unrealistic in paediatric DoC, preventing the conducting of large clinical trials, one might question whether this is an insurmountable problem. In fact, the chronic manifestation of DoC offers opportunities for dosage escalation and systematic dosage exploration *within* subjects, which can similarly contribute to the definition of standards. In instrumental trials, these include investigating multiple stimulation parameters and treatment intensities, which should always be designed depending on age and levels of neurological compromission, evaluated within the safety, and efficacy operational ranges and described in the trial protocol.

Even in cases when an adult-to-child translation of the treatment is technically feasible [104,105,106,107], it should not be aprioristically assumed to be correct. For example, tES protocols for increasing the motor evoked potential amplitudes in adults can have a paradoxical effect in children, thus decreasing those potentials when the same current intensities are applied [108] due to the different physical and geometrical properties of the skull [104].

### 6.2. Precision

In paediatric DoC, precision medicine requires rigorous clinical trial design. DoC encompasses heterogeneous aetiologies and severities, which inject variability of response into clinical trials. Inclusion criteria relying on narrower endotypes, rather than generically addressing DoC as a whole, can help obtain larger effects in smaller samples. The same result can be achieved with a more stringent selection of treatment windows and periods to evaluate (post)treatment efficacy. Biomarkers, including instrumental and “wet” markers, can similarly reduce variability by profiling the individual sensitivity to treatment, documenting therapy-induced changes, and monitoring undesired effects such as maladaptive plasticity.

### 6.3. Investments

A collaborative effort is essential to attract funds. In the USA, Europe, and worldwide, recent large-scale initiatives such as TRACK-TBI (Available online: https://tracktbinet.ucsf.edu, accessed on 30 January 2022), CENTER-TBI (https://www.center-tbi.eu, accessed on 30 January 2022), and the more recent Coma Curing Campaign (https://www.curingcoma.org/home, accessed on 30 January 2022), to mention a few, have partially addressed DoC, paediatric TBI, or both. However, resources are needed to focus specifically on the sub-acute and chronic phase of DoC and the paediatric peculiarities of the disease. Importantly, funding bodies should regard these specifications as topics deserving development and resources rather than niche areas with limited impact or generalisability. Research funding schemes should encourage centres located in middle-income countries to participate in multicentre networks and to share scientific knowledge and clinical competencies beyond on-purpose global health calls.

### 6.4. Comparability, Open-Sourcing, and Data Enrichment

Rehabilitation trials are long and expensive, with intensive use of highly trained professionals. Instrumental trials equally need human resources, have high running costs, and often require the availability of infrastructure. For these reasons, multicentric studies, comparative research, open data infrastructure, and open access policies for data collection are crucial tools to achieve the sustainability and equity of research. To inform clinical research, statistics and artificial intelligence can provide useful tools such as simulations, in silico trials, and digital twins for synthetic data generation or enrichment [109]. However, these methodological efforts need specific funds and acknowledgement.

### 6.5. Ethics

Intuitively, treatment improvement directly impacts children with DoC, and even more so in contexts where end-of-life decisions are taken in consequence of poor response to therapies. In any case, the overall higher chances of good long-term prognosis of DoC for children compared to adults should be considered [4,5]. In addition, end-of-life decisions should never be taken without a careful pain assessment, including instrumental examination. Healthcare professionals should consider that families might be traumatised or overloaded with medical information when end of life is discussed. As a result, families and caregivers might not have a complete understanding of the reasons guiding therapeutic choices and might be unprepared for the expected effects, including risks. This might bias communication, creating over- or under-expectations and having an impact on irreversible decisions. Conversely, the risks of adverse effects of treatments and maladaptive phenomena should be evaluated within the perspective of the entire lifespan. Another sensitive topic is the enrolment of placebo groups in clinical trials, which should be avoided through advanced trial designs [110]. Artificial intelligence and synthetic methods should always be used ethically, towards an improved, more focused, or more productive use of resources, and never for their calculated subtraction, reduction, or delay. Lastly, monitoring activities might be required in countries where the establishment of Ethics Committees for human research is still in process, or where boards have less oversight on research practices.

### 6.6. Strategic Investments in Low- and Middle-Income Countries (LMICSs)

Some recent virtuous initiatives in LMICs are setting a precedent for how to quickly implement strategies to fill gaps and align with high standards of care for paediatric DoC. As an example, Armenia, Georgia, and Moldova, three countries with recent historical affinity and a common healthcare heritage, have joined forces to attract funds and invest in road safety, TBI surveillance, creation of a common data registry infrastructure, post-acute service delivery, extension of access to post-acute care, and quality assessment of their healthcare, including trauma care systems [111]. They have also lobbied to obtain funds specific to education, increase the preparedness of healthcare professionals, and establish good quality CPE, including the paediatric theme (NIH/NINDS R21 NS098850). Governmental awareness and public investments have been crucial in meeting the international funders’ trust, with the ultimate achievement of reversing the otherwise increasing TBI rates [111].

In 2017, the World Health Organization (WHO) issued a call for action to draw attention to the increasing demand for rehabilitation worldwide, promote rehabilitation in health care systems, and support the availability and accessibility of rehabilitation and assistive technologies to achieve “healthy lives and wellbeing for all, at all ages” (https://www.who.int/disabilities/care/Rehab2030MeetingReport_plain_text_version.pdf; https://www.who.int/news-room/fact-sheets/detail/assistive-technology, accessed on 30 January 2022). One year later, the International Paediatric Brain Injury Society and The Eden Dora Trust created a toolbox to assist and guide professionals involved in the rehabilitation of children, adolescents and young adults with brain injury (https://www.ipbis.org/toolboxupdate2021.html, accessed on 30 January 2022). We believe that these initiatives, flourishing thanks to increasing awareness of the importance of DoC care and post-acute rehabilitation, point to a participative and global future for paediatric DoC research.

## 7. Conclusions

In conclusion, the consciousness outcome of children with DoC after injury is generally more optimistic than observations in adults; survival is long, and late emergence is possible, which call for investment in treatment and extreme caution in withdrawal of care and end-of-life decisions. The overall advancement in the field of paediatric DoC ultimately depends on global awareness of the increasing demand for post-acute care, specific professional training in paediatric DoC treatment, and focused generation of funds for paediatric DoC.

## Figures and Tables

**Figure 1 brainsci-12-00198-f001:**
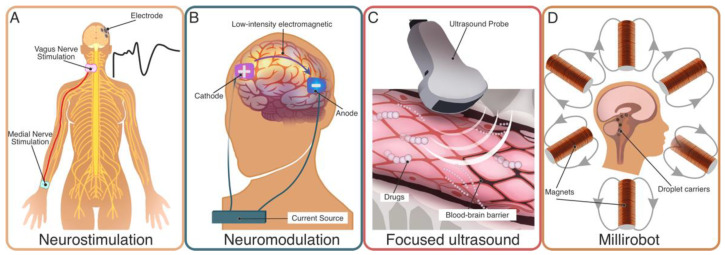
Medical technologies for treatment of Disorders of Consciousness. From left to right: (**A**) neurostimulation, an electrical stimulation to a primary nerve afferent to the brainstem and cortex; (**B**) neuromodulation: a low-intensity electromagnetic treatment delivered to the cortex; (**C**) focused ultrasound: transient, localised, and graded opening of the blood–brain barrier using ultrasounds; (**D**) millirobot: controlled drug delivery to target tissues through soft droplet carriers manipulated by external magnetic fields.

**Table 1 brainsci-12-00198-t001:** Disorders of consciousness—definitions (from Edlow et al. [7]) and available epidemiology (overall DoC prevalence: ~0.2 to 3.4 per 100,000 individuals in Europe [2]).

State	Definition	Available Epidemiology
Coma	Complete absence of arousal and awareness	Incidence of non-traumatic coma in UK: 30.8/100,000 children under 16 per year; 6.0/100,000 of general population per year [8].
Vegetative State/Unresponsive Wakefulness Syndrome (VS/UWS; formerly also Apallic State)	Arousal without awareness	Incidence: ~2.6/100,000 people [9]. Prevalence: ~2.0/100,000 to ~5.0/100,000 people, depending on national protocols [9].
Minimally conscious state minus (MCS-)	Minimal, reproducible, but inconsistent awareness without language	Prevalence: ~2.2/100,000 in Europe [2].
Minimally conscious state plus (MCS+)	Minimal, reproducible, but inconsistent awareness with language comprehension and expression (i.e., either command following, intelligible verbalization or intentional communication).	
Emergence to consciousness (eMCS), including the Confused-Agitated State (CAS)	Persistent dysfunction across multiple cognitive domains, behavioural dysregulation, disorientation, also with symptom fluctuation.	Indirectly estimated in: ~0.4/100,000 in a single centre study in Europe [10].
Cognitive Motor Dissociation (CMD) *	Volitional brain activity with no behavioural manifestation.	Unknown.

* Recently introduced.

**Table 2 brainsci-12-00198-t002:** Exclusion criteria for non-invasive brain stimulation, brain modulation, and drug delivery.

Treatments	Exclusion Criteria
Neurostimulation	(Limiting, although not excluding) Inability to communicate pain.
Neuromodulation	Presence of epilepsy [94], unless the intervention is specifically performed to treat this complication. This applies to tES in general. However, higher associated risk of inducing seizures [95] is reported for repetitive transcranial magnetic stimulation (rTMS). Presence of subclinical seizures (to be ascertained with a neurophysiological examination). Sedative drugs, NMDA receptor antagonists, and Na+ or Ca++ channel blockers, which might cause (unplanned) interaction with the modulatory effect generated by the electrical currents or magnetic fields. Metal implants [47]. (Limiting, although not excluding) Presence of multiple (focal) lesions, such as in the case of traumatic brain injury, which cause the targets to be multiple or not identifiable.
Drug delivery	Allergy to chemical vectors [88]. Certain inaccessible location of the anatomical structure to be targeted by drug delivery.

## Data Availability

Not applicable.

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
