# Peer review of "Emerging Treatments for Disorders of Consciousness in Paediatric Age"

_brainsci, 2022, doi:10.3390/brainsci12020198_

Round 1

Reviewer 1 Report

Review brain sciences

Thank you for the great honor and pleasure to be invited to review the manuscript:

Emerging treatments for disorders of consciousness in paediatric age

General comment:

The authors must be congratulated for this valuable commentary that attempts to describe the current state regarding treatments of DoC in children and adolescents.  

It is a hugely important topic as the knowledge on this is still rather scarce; this concise yet comprehensive commentary is certainly worth publishing.

This review will be short as this article reads very well and covers important considerations not only in the treatment but also in the prevention of DoC – and should not be delayed in publishing.

Specific (minor) comments:

Line 73/74: the authors describe the treatment options of reduction of DoC related complications and mention spasticity. In the daily clinical work in the rehabilitation of DoC patients, the treatment of spasticity and dystonia (and related prophylaxis of contractures, deformities) plays a major role. This reviewer would like to see more than a bracket dedicated to this topic, especially as the fatigue-inducing side effects of these medications for treatment of spasticity/dystonia often hampers the intensions to improve consciousness. Are there new/specific recommendations (or not)?

Line 102/103: would it be possible to be more specific regarding the meaning of “early after the injury”?

How about recovery of sleep in children with DoC?
Are there any new/specific recommendations?

Author Response

Dear Reviewer 1,

PLEASE SEE ATTACHED FILE

With reference to your correspondence of 18 January 2022, we thank you for the quick feedback, and we are pleased to hear that our manuscript was well-received and considered for publication in Brain Sciences.

Please find attached a revised version of our manuscript, with tracked changes. Also, we detail below the point-by-point reply to your comments and remarks.

We are grateful for the fast turnaround, and we hope that the additional changes made at this stage are satisfactory. We remain available to further refine the manuscript, should this be needed.

Also, we remain available to the Editorial staff for clarification and in case additional adjustments are required. We look forward to hearing from the Editorial Office.

Yours sincerely

Erika Molteni and Hassna Irzan, on behalf of all the authors

Reviewer 2 Report

Thank you for the opportunity to review this paper. I found it interesting to read and I think it is written in a style which is easily accessible, including for researchers and clinicians who have an interest in the topic but may not specialise in this field. 

Introduction
I would suggest adding a sentence somewhere in the Introduction explaining any differences in outcomes for paediatric and adult populations as later in the review you make mention of this.

Lines 50-2: I suggest splitting this into two sentences to aid readability:
"However, diverse socio-cultural factors, health policies, and hygiene standards cause both the incidence and prevalence of DoC to fluctuate between countries [add reference - 2?]. Furthermore, early withdrawal of care may contribute reduced the prevalence of DoC in some countries [2]."

Lines 67-72
It wasn't clear to me why you discuss the measuring of DoC and tools currently used over several sentences as it didn't seem to be discussed again later as an aspect which might influence treatment (although logically how you measure DoC would impact on measuring treatment efficacy). I would suggest clarifying why this is discussed in the Introduction as it didn't appear to be a key point raised again later in the paper.

Lines 73-79
1. In lines 74-5 you mention that amantidine is the only therapy recommended in clinical guidelines but in line 77 you say that there are no guidelines for treatment of paediatric DoC. Do you mean that the former are clinical guidelines for adults? I would suggest clarifying.
2. Do you mean that the only therapy recommended for consciousness in line 74 or for DoC complications?
3. Please add a reference for amantidine being only effective in about 55% of cases

Table 1
Note for the typesetters (or the authors if they can adjust the table layout):
A space is needed before 'Prevalence: ~2.2/100000 [2].
In Europe.' to separate this text from the text in the previous column

For the authors:
Row 1: suggest moving 'In United Kingdom.' higher so it reads: 'Incidence of non-traumatic coma in the United Kingdom'
Row 3: amend so it reads 'Prevalence: ~2.2/100000
in Europe [2].' (to match how you present information in Row 4)
Row 4: Please add a reference for Estimated in: ~0.4/100000 in a single centre study in Europe.

Line 88-9 - amend 'aims to' to 'aims for'

Lines 88-96: Please add references for information in paragraph of lines 88-96
Lines 97-8: Please add a reference in lines 97-8 for: '(Multi)sensory stimulation is at the core of early intervention. It aims at generating a comprehensive set of environmental prompts through stimulation of all the five 98 senses and the motor system.'
Lines 97, 102 and 106: bold subheadings - question words. I would suggest expanding these question words to a full sentence as you have done on line 88 as it reads better (and is consistent with your next section's subheadings)

Lines 104-5 - amend 'who achieve' to 'which achieve'

Lines 109-10 please add reference for "and themes matching the pre-injury hobbies and inclinations have been proven effective."

Lines 117-8: add 'elements' after 'these' so it reads 'these elements to improve'

Lines 163-5: please add reference

Line 179: suggest amending 'confounder' to 'a confounding variable' or similar

Line 207: amend 'Why to use technology?' to 'Why use technology?'

Line 210: delete 'So,'

Lines 213-4: please check a word isn't missing in: "Sensorimotor stimulation specifically candidates to be delivered through devices."

Lines 227-8: should this say 'in reported cases'?

Line 230: amend 'to improve' to 'of improving'. Also, please check whether the use of 'proven' (which states a very high degree of certainty) is appropriate to describe the results for this study. 

Line 234: I think 'alternate' is possibly not the correct adjective here as it means that every other case would be successful according to a regular pattern

Lines 249, 266, 275: I suggest expanding the question word subtitles to a sentence so it is clear what the question is

Line 262: I suggest adding a reference for the NCCPC-PV

Line 264: delete 'do'

Lines 266-74: are there any references which could be cited for some of the information in this paragraph?

Figure 1: Consider placing this before subsection 4.2

Table 2: I suggest putting the bolded headings in column 1 and the criteria in column 2
As Table 1 included references, it may be good to add some to Table 2 for key points (you have added references for some points but not all)

Section 5. The way forward - this reads as a conclusion section to the entire review but then does not discuss the points raised with reference to LMIC. Is there scope in Section 5 to make any suggestions for LMIC? I felt the recommendations and suggestions in this section didn't really touch on how they could be translated more broadly to LMIC (e.g. is some of the technology even available in low-income countries? How could it be made available? How would ethics applications differ? Are there safeguards required in countries where there is less oversight by ethics boards for human research? What role in research would different mechanisms of DoC play which may differ between LMIC and high-income countries (e.g. have differences in DoC caused by TBI vs infectious diseases [which you say in the introduction are more prevalent in low-income countries] been examined and would this impact treatments?). I think adding some considerations for LMIC would add a lot of value and broaden the research focus from focusing on just high-income countries. If this information was added, I would suggest reordering sections 5 and 6. 
If this information can't be added, then I would amend the title of section 5 to something which restricts the scope of the title (but that I mean 'The way forward' suggests an all encompassing section, whereas a title like 'Considerations for future research into therapies for paediatric DoC' is more specific). 

Line 344: Please add references to support the sentence that children with DoC have better recovery than adults with DoC and clarify what you mean by 'prognosis' - for example do you mean regaining consciousness or do you mean recovery outcomes from brain injury?  In the TBI literature, some researchers have suggested that children have poorer recovery than adults as they can fail to achieve the same developmental trajectories as their peers (e.g. in development of executive function skills). Also, this seems to contradict the statement in the Introduction that outcomes are poor (lines 56-7). 

Lines 342-352: this is just a suggestion but I feel families should be mentioned in this paragraph. For example treatment improvement impacts children with DoC and their families and communicating risks associated with treatment at a time when families are traumatised and overloaded with medical information can be challenging to gaining appropriately informed consent (e.g. families may agree to treatment because they overestimate the benefits etc). It is useful to remind researchers of these broader considerations.

Lines 356-9. I suggest adding some semi-colons so the causes are better separated (see my example below). Also could you clarify:
lower safety standards - do you mean on roads?
older or insufficient infrastructures - do you mean medical infrastructure?
governmental neglect - is this with reference to medical resources, road maintenance or something else?

"Common causes are poorer road conditions and lower safety standards; high rate of infections and cerebral malaria in children who tend to be malnourished; older or insufficient infrastructures; war conflicts; governmental neglect; and urban-centred services leaving rural or remote areas underserved."

Line 361: 'due to poorer childminding culture and home supervision' - this reads that parents in LMIC are not as good at caring for their children and monitoring their children's welfare which I don't think is the intention. Are there perhaps other factors which need to mentioned e.g. socioeconomic conditions which shape parenting (increased poverty/war/disease means parents are not available to parent; children work in dangerous conditions, low parenting knowledge around TBI, etc.)

Line 371: 'poorly quality-assessed' - this wasn't clear to me. Do you mean 'poor quality'?
Line 377: please clarify 'religious driver' - do you mean strong religious beliefs? Or strong religious communities?
Line 379: amend 'access of students' to 'student access'
Line 383: "former Soviet Republics," - this is a minor point but would it perhaps be better to define these republics by geography? For example, it is unclear if this means former republics in the Baltic region, Central Asia, etc. 

Conclusion: 
Lines 405-6: As I mentioned in an earlier comment, I suggest specifying what you mean by 'outcome for children with DoC'

General editorial point:
I noticed the paper uses both spellings with -ise and -ize. I would check with the journal style and standardise this spelling
Check use of punctuation after subtitles (e.g. see 5.1 which has a full stop and 5.2 which doesn't)

Author Response

Dear Reviewer 2,

PLEASE SEE ATTACHED FILE

With reference to your correspondence of 23 January 2022, we thank you for the quick feedback, and we are pleased to hear that our manuscript was well-received and considered for publication in Brain Sciences.

Please find attached a revised version of our manuscript, with tracked changes. Also, we detail below the point-by-point reply to your comments and remarks.

We are grateful for the fast turnaround, and we hope that the additional changes made at this stage are satisfactory. We remain available to further refine the manuscript, should this be needed.

Also, we remain available to the Editorial staff for clarification and in case additional editorial/typesetting adjustments are required. We look forward to hearing from the Editorial Office.

Yours sincerely

Erika Molteni and Hassna Irzan, on behalf of all the authors
